# Optimization of Sample Preparation for Metabolomics Exploration of Urine, Feces, Blood and Saliva in Humans Using Combined NMR and UHPLC-HRMS Platforms

**DOI:** 10.3390/molecules26144111

**Published:** 2021-07-06

**Authors:** Cécile Martias, Nadine Baroukh, Sylvie Mavel, Hélène Blasco, Antoine Lefèvre, Léa Roch, Frédéric Montigny, Julie Gatien, Laurent Schibler, Diane Dufour-Rainfray, Lydie Nadal-Desbarats, Patrick Emond

**Affiliations:** 1UMR 1253, iBrain, University of Tours, Inserm, 37044 Tours, France; cecile.martias@univ-tours.fr (C.M.); nadine.baroukh@univ-tours.fr (N.B.); sylvie.mavel@univ-tours.fr (S.M.); helene.blasco@univ-tours.fr (H.B.); antoine.lefevre@univ-tours.fr (A.L.); frederic.montigny@univ-tours.fr (F.M.); diane.dufour@univ-tours.fr (D.D.-R.); patrick.emond@univ-tours.fr (P.E.); 2CHRU Tours, Medical Biology Center, 37000 Tours, France; 3ALLICE, Phenotyping Station, 37380 Nouzilly, France; lroch8@gmail.com (L.R.); julie.gatien@allice.fr (J.G.); laurent.schibler@allice.fr (L.S.)

**Keywords:** biological fluids, extraction protocol, UHPLC-HRMS, ^1^H-NMR, metabolomic card

## Abstract

Currently, most clinical studies in metabolomics only consider a single type of sample such as urine, plasma, or feces and use a single analytical platform, either NMR or MS. Although some studies have already investigated metabolomics data from multiple fluids, the information is limited to a unique analytical platform. On the other hand, clinical studies investigating the human metabolome that combine multi-analytical platforms have focused on a single biofluid. Combining data from multiple sample types for one patient using a multimodal analytical approach (NMR and MS) should extend the metabolome coverage. Pre-analytical and analytical phases are time consuming. These steps need to be improved in order to move into clinical studies that deal with a large number of patient samples. Our study describes a standard operating procedure for biological specimens (urine, blood, saliva, and feces) using multiple platforms (^1^H-NMR, RP-UHPLC-MS, and HILIC-UHPLC-MS). Each sample type follows a unique sample preparation procedure for analysis on a multi-platform basis. Our method was evaluated for its robustness and was able to generate a representative metabolic map.

## 1. Introduction

The metabolome is considered to represent the ultimate endpoint of the biological cascade (genomic, transcriptomic, proteomic). Impacted by the complex interactions between the host and its microbiota and other factors such as diet, stress, gender, age, and lifestyle [1,2], the metabolome reflects the biological phenotype of an individual and is very often targeted in life science research to capture metabolic-induced changes in physiopathological conditions to highlight dysregulated biological pathways. Metabolomics focuses on the wide variety of low molecular weight metabolites (<1500 Da) present in biological samples (cells, tissues, or biological fluids). Metabolomics has become a popular approach for metabolite biomarker discovery by comparing the metabolic profiles from different phenotypes in order to identify specific metabolic changes, leading to the understanding of physiology, toxicology, and disease progression. In the last decade, a lot of data have been generated using metabolomics, resulting in a better understanding of many biological systems [3,4] and diseases such as diabetes [5,6,7], Crohn’s disease and inflammatory bowel disease [8,9,10], cancer [11,12,13], cardiovascular disease [4], atherosclerosis [14], obesity [15,16,17], and psychiatric disorders [18].

Metabolomics implies a specific workflow [19] starting from the pre-analytical work (sample collection, sample pre-processing as metabolic quenching, centrifugation, filtration, and sample storage) followed by analytical work (sample processing, pre-acquisition normalization, data acquisition (NMR, GC-MS, LC-MS)) and ending with data processing (data pre-processing, statistical analysis, metabolite identification, metabolite quantification) to identify biomarkers and/or to understand biological dysregulations [20]. Each step of this workflow is crucial for identifying reliable biomarkers to be transferable in clinical studies. In addition, metabolomics combined with bioinformatics methods open a path for large-scale data analysis and give access to the analyses of associated metabolic pathways.

Currently, most of the clinical metabolomics studies are completed on a single biological matrix (either blood or urine, or any other biological fluids or tissues) or using one analytical platform, either nuclear magnetic resonance (NMR) or chromatography–mass spectrometry (MS), which is one of the most widely applied technologies. In the literature, blood (serum or plasma) is described as one of the most frequently used biological materials in clinical metabolomics research. Blood (serum and/or plasma) moves many metabolites through the body and may provide a lot of metabolite information about physiological and pathophysiological conditions in a particular biological system [21]. Urine gives another alternative to blood, and it is necessary for metabolic interpretation. Urine contains mainly metabolic breakdown products from foods, drinks, drugs, environmental contaminants, endogenous waste metabolites, and bacterial by-products [21], which makes it ideal for early detection of a wide range of disease [22,23]. Urine may be used as a reflection of the endogenous and exogenous metabolic profiles at a precise moment.

The use of fecal samples in metabolomics is increasing. This growth is driven by the hypothesis that the fecal metabolome may capture the complex interactions between the gut microbiome, the host, and the diet [24]. The role of gut microbiota has been correlated with multiple gastrointestinal diseases, obesity [25], rheumatoid arthritis [26], psychiatric diseases such as autism [25] or depression [25], cardiovascular diseases, and type 2 diabetes [6]. The vast majority of clinical studies tend to focus on a single biological matrix, even if combining data from various matrices should reveal interesting correlations across compartments. From this situation, one could wonder how to describe as closely as possible a human metabolome with only one or two biofluids if we consider the biological complexity and systemic regulation between different biological compartments (blood and digestive or urinary tracts).

Owing to the complexity and size of the human metabolome and the diverse physical and chemical properties of metabolites, it appears that no single analytical platform can be used to screen all metabolites in a biological sample [27].

Another concern is sample preparation, in the literature, different procedures were proposed depending on the biofluids [28,29,30,31] and on the analytical techniques used [28,32,33,34]. Depending on the kind of molecules sought, foreseeable consequences would be based on the laboratory results. Currently, there is no consensus, no standard operating procedure (SOP) to prepare biological samples. This lack of SOP generates pitfalls when comparing inter-lab studies. Actually, metabolomics is very much research laboratory-based and needs to move out of academic laboratories into clinical routine.

Although, some studies have already investigated metabolomics data from multiple fluids [24], the information has stayed limited to a unique analytical platform. In contrast, clinical studies investigating the human metabolome that combine multi-analytical platforms have focused on a single biofluid [35]. Combining data from various matrices captured with NMR and UHPLC-MS platforms would permit for the enhancement of the metabolic compound coverage. This proof of concept would help to reveal potential new biomarkers in clinical contexts with correlations across matrices.

Currently, there are no study combining multiple matrices and multiple platforms. Thus, the present study describes a standard operating procedure for four biological specimens (urine, blood saliva, and feces) using multiple platforms (NMR, UHPLC(RP)-MS, and UHPLC(HILIC)-MS). Here, we aimed to find a unique sample preparation procedure for each biological sample type to be use on multiple platforms and achieve the widest metabolite coverage to trace a large number of metabolic pathways. Combining multiple analytical methods with several numbers of biofluids should provide a better characterization of a metabolic human global profile and should enable us to illustrate a patient’s metabolic card.

## 2. Results and Discussion

The global analytical workflow applied for each biological sample to select the optimal preparation protocol is shown in Figure 1.

### 2.1. Optimization of Sample Preparations

In this study, we aimed to obtain a single preparation procedure that can be used to explore the metabolome by NMR, ESI+ and ESI- RP-LC-MS, and ESI+ and ESI- HILIC-LC-MS platforms, given the better metabolic coverage in term of number of metabolites detectable and the highest reproducibility (CV%) of polar metabolites.

#### 2.1.1. Urine

We focused on comparing the efficiency of different preparation protocols (see Appendix A) such as monophasic extraction with two different solvents (MeOH or ACN) or dilution preparation (1:2 for NMR and 1:10 for LC-MS) as has already been described in the literature [28,36]. Each preparation protocol was able to provide a large number of metabolites, as shown in Figure 2a. Untargeted data are provided in the Appendix A for guidance. Indeed, we focused on biological information made up of compounds identified at the level 1 of confidence, which is a minimum prerequisite to be used for a biochemical interpretation in a clinical routine. However, these untargeted data did not allow us to assess the redundancy of the metabolic information between analytical platforms (see Appendix A). According to the data-fusion step, the chemical redundancy was removed, and considering all platforms, we detected over 176 compounds with the preparation of urine/ACN (1:8), among which 145 metabolites had a CV < 30%. We found over 215 compounds with the preparation urine/MeOH (1:8), from which 201 metabolites had a CV < 30%. With the dilution protocol of over 205 compounds, 197 metabolites showed a CV < 30% (Appendix A). The comparison of the two monophasic preparations of MeOH versus ACN, indicated that MeOH was the more appropriate one to use. For MeOH monophasic preparation and the dilution preparation, the results were similar. According to the CV < 10%, the dilution had a better reproducibility, showing 73% of CV < 10%, while urine/MeOH (1:8) had 62.2% of CV < 10%. Regarding the number of compounds with CV < 30%, 131 were accessible with the three preparations and, interestingly, some of them were found to be unique to a specific preparation. We detected 22 unique compounds associated with the urine/MeOH (1:8) preparation, 19 with the urine/H_2_O preparation, while only 5 unique compounds were assessed with the urine/ACN (1:8) preparation (Figure 2a). This result indicated that urine/ACN (1:8) provided minor compound diversity.

In order to select the most relevant preparation procedure for urine sample optimization, we looked at the metabolomics data processing level using the Network Explorer module in MetaboAnalyst 4.0 and found additional important biological information. Metabolic pathways were then bioinformatically predicted for each set of metabolites (see Appendix A). Results indicated that the urine/MeOH (1:8) preparation had the most exhaustive metabolic coverage with 12 metabolic pathways described.

In term of the total number of metabolites extracted, reproducibility, and metabolic pathway description, the urine/MeOH (1:8) preparation with 201 chemicals extracted (CV < 30%) including 22 unique compounds had the largest coverage of the metabolite sets.

#### 2.1.2. Blood

A total of six preparation protocols were tested with blood samples (see Appendix A): four monophasic extractions using two different solvents, ACN or MeOH within 1:2 and 1:8 proportions, and two biphasic extractions, blood/MeOH/CHCl_3_ at (1:1:1) and (1:1.5:2.5) ratios. All extraction protocols succeeded in providing a wide number of metabolites, as shown in Figure 2b. As mentioned in the urine section, the untargeted data are shown in the Appendix A. Once the chemical redundancy was eliminated, the monophasic extraction protocols detected over 139 compounds on all platforms, with 129 metabolites showing a CV < 30% in the blood/ACN (1:2) preparation. In the blood/ACN (1:8) preparation, over 135 compounds were detected on all platforms, 123 metabolites had a CV < 30%. In the blood/MeOH (1:2) preparation, over 147 compounds were detected on all platforms, 141 metabolites had a CV < 30%. With the blood/MeOH (1:8) preparation, over 157 compounds were found on all platforms, 143 metabolites had a CV < 30%. The biphasic protocols extracted over 127 compounds detectable on all platforms, 119 metabolites had a CV < 30% with the (blood/MeOH/CHCl_3_) (1:1:1) platform, while over 126 compounds were detectable on all platforms and 118 metabolites had a CV < 30% with the (blood/MeOH/CHCl_3_) (1:1.5:2.5) (see Appendix A). Comparing the six preparations, results were similar in term of numbers of metabolites. According to the CV < 30% (Figure 2b), methanol precipitations (1:2) or (1:8) had the best metabolite recoveries with, respectively, 141 and 143 metabolites, while acetonitrile precipitations (1:2) and (1:8) had, respectively, 129 and 123 metabolites. The biphasic extraction (blood/MeOH/CHCl_3_) (1:1:1) had 119 metabolites with a CV < 30% while (blood/MeOH/CHCl_3_) (1:1.5:2.5) had 118 metabolites with a CV < 30%. In terms of reproducibility, 53.2% of CV were <10% with blood/MeOH (1:2), while 75.3% of CV were <10% with blood/MeOH (1:8). Regarding the number of unique compounds for each preparation (Figure 2b), only one compound was unique for blood/ACN (1:8) and blood/MeOH (1:2), and two were unique for blood/ACN (1:2). The blood/MeOH (1:8) preparation had 10 unique compounds, meaning that this preparation provided the most information in terms of compound diversity. The biphasic extractions provided none and two unique compounds, respectively, for the (1:1:1) and (1:1.5:2.5) extraction conditions.

Similar to the process for urine, Network Explorer was used to estimate metabolic pathways that might be involved (see Appendix A). Each preparation covered nine metabolite sets (*p*-value < 0.05) with six or seven pathways with a *p*-value < 0.01 depending on the preparation. As we can see, few differences was found.

In terms of the number of metabolites extracted, reproducibility, and metabolic pathway description, the blood/MeOH (1:8) preparation with 143 chemicals extracted (CV < 30%) including 10 unique compounds had the best metabolite coverage. The blood/MeOH (1:8) preparation was the most suitable preparation.

#### 2.1.3. Saliva

Saliva preparations were tested with four preparation protocols (see Appendix A). All preparations were able to provide wide metabolite coverage, as shown in Figure 2c. As mentioned in the urine section, the untargeted data are shown in the Appendix A. Once the chemical redundancy was eliminated, the preparations saliva/ACN (1:1) and saliva/ACN (1:2) detected a very similar number of metabolites; over 148 compounds or 147 were detectable on all platforms, with 137 metabolites and 136 having a CV < 30% with the preparations saliva/ACN (1:1) and saliva/ACN (1:2), respectively (see Appendix A). With the preparations saliva/MeOH (1:1) and saliva/MeOH (1:2), 145 and 146 compounds were detectable on all platforms, respectively, with 133 metabolites having a CV < 30% with the two methanol preparations. Comparing the four preparation precipitations in term of numbers of metabolites, results were similar. In term of reproducibility, 48.9% of CV were < 10% with saliva/ACN (1:1) while 56.6% of CV were <10% with saliva/ACN (1:2). Regarding the number of unique compounds for each preparation (Figure 2c), four unique compounds were found with the saliva/ACN (1:1) preparation and two unique compounds were found with the saliva/ACN (1:2), saliva/MeOH (1:1), and saliva/MeOH (1:2) preparations.

Once again, Network Explorer was used to predict metabolic pathways (see Appendix A). Each preparation studied allowed us to cover eight metabolite sets (*p*-value < 0.01) except the saliva/MeOH (1:2) preparation, which showed nine metabolite sets, as shown in Appendix A. 

In terms of the number of metabolites extracted, reproducibility, and metabolic pathway description, the saliva/ACN (1:2) preparation with 136 chemicals extracted (CV < 30%) including two unique compounds was the most suitable preparation method for saliva samples.

#### 2.1.4. Feces

Six extraction protocols (see Appendix A) were tested on freeze-dried feces, (ACN/H_2_O, MeOH/H_2_O) with two ratios (1:1 and 4:1), and (MeOH/ACN/H_2_O) (1:1:1) and H_2_O (Figure 2d).

The six extraction protocols were able to provide wide metabolite coverage, as shown in Figure 2d. As mentioned in the urine section, the untargeted data are shown in the Appendix A. Once the chemical redundancy was eliminated, the MeOH/H_2_O (4:1 and 1:1) preparations had a total of 270 and 278 compounds with 259 and 264 molecules with a CV < 30%, respectively. The ACN/H_2_O (4:1 and 1:1) preparations had a total of 272 and 281 compounds with 259 and 266 molecules with a CV < 30% respectively. The MeOH/ACN/H_2_O and H_2_O preparations had a total of 275 and 269 compounds with 258 and 254 molecules with a CV < 30%, respectively. In terms of the number of compounds (Appendix A), MeOH/H_2_O and ACN/H_2_O ratio (1:1) preparations seemed to be the most effective ones. Comparing these two preparations in term of numbers of metabolites with CV < 30%, results were quite similar. In term of reproducibility ACN/H_2_O (1:1) had 60.5% of CV < 10% and MeOH/H_2_O (1:1) had 72% of metabolites with CV < 10%. Regarding the number of unique compounds for each preparation (Figure 2d), eight compounds were found only with MeOH/H_2_O (1:1), while only three were unique with ACN/H_2_O (1:1).

Network Explorer was used again (see Appendix A). Each preparation covered between 10 and 12 metabolite sets (*p*-value < 0.01), as shown in Appendix A.

In term of the number of metabolites extracted, reproducibility, and metabolic pathway description, the MeOH/H_2_O (1:1) preparation with 264 chemicals extracted (CV < 30%) including 8 unique compounds was the most suitable method preparation for urine samples.

### 2.2. Overall Synthesis on Optimization of Sample Preparation

The present study focused on urine, blood, saliva, and feces preparations, which is a critical point in global metabolic profiling, to be analyzed by combined NMR and UHPLC-HRMS platforms in order to describe an individual from their different compartments.

According to the literature, urine is usually diluted in water [36] or with buffered solution [37] depending on the analytical tool used. NMR needs a pH adjustment before acquisition to avoid chemical-shift bias in the automated analysis. Using NMR, it is difficult to quantify metabolites if some macromolecules such as proteins are still present in the sample because of the baseline distortion. UHPLC-MS is more sensitive than is NMR and metabolites are first separated by liquid chromatography before the analysis by mass spectrometry. Comparing solvent precipitation (urine/methanol (1:8)) to the dilution preparation (quick and simple), we obtained as many metabolites as in the diluted preparation with good reproducibility, and the NMR spectra did not need realignment and the baselines were not distorted, allowing for quantification if needed. In our study, combining NMR with UHPLC-MS, urine/methanol (1:8) precipitation gives the most rapid, reproducible metabolomics information and allowed for good metabolic pathway coverage. 

Several papers showed that blood is usually prepared with a protein precipitation using acetonitrile or methanol [24,31,36] or a methanol/chloroform precipitation [38] for macromolecule removal. SPE, hybrid-SPE, or micro-extraction [31] can be used, but those methods are time-consuming and costly, when considering large cohort exploration. In our study, using NMR with UHPLC-MS, blood/methanol (1:8) precipitation gives the most rapid, reproducible metabolomics information and allowed for good metabolic pathway coverage. 

In the literature, saliva usually is centrifuged and either diluted with a buffered solution for NMR exploration [39] or used with a methanol protein precipitation in LC-MS. Recently, in a targeted metabolomics study using UPLC-MS/MS on the saliva of children, Schultz et al. [40] removed mucins and bacterial debris by centrifugation and proteins by acetonitrile-mediated precipitation. In our study, combining NMR with UHPLC-MS, the preparation saliva/acetonitrile (1:2) was the most suitable sample preparation.

In the literature, feces is described as a complex matrix with very diversified sample preparations [41]. For NMR exploration, the most commonly used extracting solvent is phosphate buffered saline or distilled water [42,43]. For LC-MS analysis, fecal water [34] or lyophilized feces are dissolved in ultrapure water before solvent precipitation [36,44]. Moosmang et al. [45] were the first to establish a protocol for the extraction of human fecal samples for NMR and LC-MS techniques. In their study, they concluded that water was the solvent of choice for fecal sample extraction. In our study with lyophilized feces, methanol/water (1:1) as the extraction solvent yields better results in terms of the number of detected metabolites as well as reproducibility.

### 2.3. Intraday Precision

According to the optimization of sample preparations, the intraday precision was evaluated. Repeated samples (n = 16) were prepared with the chosen extraction preparation and the variation coefficients of metabolites were calculated (Table 1).

We aimed to validate the selected preparation for each biological compartment by an intraday precision. For the optimization of the sample preparation method, a CV < 30% was considered acceptable regarding the analytical variability. In our validation step, only three metabolites had a CV higher than 30% in blood and urine and five for the feces matrix. All the saliva metabolites had a CV under 30%.

Since all repeated samples used during the validation step should have the same composition, the variability of their signal was only due to the analytical source. Therefore, metabolites should be present in all 16 repeated measures to be kept. The difference was due to the data processing, which was drastic and showed more consequences when the number of repetitions was larger. Indeed, the probability of deleting a metabolite that has a zero-fill among the 16 repeated measures was greater than that in the extraction samples (n = 5). Analytically, metabolites with CV below 30% were the most robust and should be considered to metabolically map an individual.

### 2.4. Platform Complementarity

The complementarity of the analytically different platforms was assessed through a Venn diagram (see Appendix A) and using selected preparations for each matrix. Dealing with the four matrices, NMR allowed us to detect between 5% and 13% of the metabolic information, HILIC and RP provided 29–52% and 20%–38%, respectively. Finally, for each biological sample, 75.6% (67%–82%) of the metabolic information was platform specific. UHPLC-MS provided more information than did NMR because this analytical technique is more sensitive than NMR but enables a quantitative aspect. The multiplatform approach is time-consuming but increases the numbers of metabolites, accuracy, and reliably. Metabolites could have a different CV according to the analytical tool used, so the best CV should be kept. Human cohorts are considerable in size, consequently, using a unique sample preparation for NMR and UHPLC-MS analysis would save time.

### 2.5. Matrix Complementarity, Metabolic Coverage, and Mapping

The complementarity of the four matrices was assessed through an upset diagram (Figure 3), which provides an efficient way to visualize intersections of multiple sets compared to the traditional approach of a Venn diagram. Among the three visualization modes of intersections [46], we decided to represent the upset plot with the degree mode. In Figure 3, the horizontal bar chart shows the entire metabolite size of each sample set (90 in saliva, 104 in blood, 151 in urine, and 208 in feces). The fecal metabolome provides the highest specificity, due to the impact of the microbiota metabolism. Degree one contains only exclusive intersections (metabolites) with exactly one participating set (sample type) without overlapping. The vertical bar chart shows the number of unique metabolite sets (66 metabolites in feces, 25 in urine, 8 in saliva, and 6 in blood). Appendix A shows the detailed list of metabolites detected in each intersection of the upset. The unique compounds reflect the complementarity of the metabolic coverage provided by each matrix. Degree two contains metabolites shared exclusively by two sets (sample type). With 31 metabolites shared, urine and feces gave the higher score of intersection. This finding is surprising given the expected correlation between urine and blood compared to urine and feces. Metabolites in blood have a short half-life because of the rapid glomerular filtration. Yet, metabolites accumulated in urine, leading to their increased concentration.

Degree three provides metabolites shared exclusively by three sets (sample type). By considering three matrices, the intersection with urine, feces, and blood gives the maximum metabolites shared (26 metabolites), which seems logical because saliva is secreted by salivary glands, which are not directly link to the other compartments.

Degree four provides metabolites shared exclusively by the four sets (sample type). In all the four matrices, 37 identical metabolites were detected. These 37 common compounds reflect the overlapping matrices and can be very informative for metabolic and physiopathological flux analysis at the organism level and to evaluate the biological compartment exchanges. These observations need to be confirmed in future studies by applying this methodology to a unique individual (we worked here with pooled samples).

While matrices complementarity and metabolic coverage was assessed earlier, metabolic mapping needs to be ascertained. iPath3.0 provides data mapping capabilities. Based on the metabolites recovered from urine, blood, saliva, and feces, we established a global metabolic card to metabolically illustrate an individual (Figure 4). This card was obtained by relocating metabolites from the intraday precision step (see Table 1). The gradient blue dots represent metabolites found in 1, 2, 3, or 4 matrices (blood, urine, saliva, feces). Considering lists of unique metabolites in each matrix described in Figure 3, the pathways described by a single matrix are for energy metabolism and metabolism of other amino acids. The metabolites common to the four matrices mainly identified nucleotide metabolism, metabolism of cofactors and vitamins, and amino acid metabolism.

While lipid metabolism and carbohydrate metabolism were not well described, this discrepancy can be explained by the sample preparation extracting polar metabolites only.

In summary, this study presents the first metabolomics fingerprinting strategy for multimatrix (urine, blood serum, saliva, and feces) analysis using a multiplatform approach ((NMR(^1^H), RP-UHPLC-MS, and HILIC-UHPLC-MS). We validated a unique preparation for each matrix exploitable with these analytical tools. This targeted multiplatform approach allowed us to detect 201, 143, 136, and 264 metabolites in urine, blood, saliva, and feces, respectively. Currently, in the literature, metabolomics studies of these biological fluids are carried out in untargeted ways [31,37,44]. In 2018, De Paepe et al. [36] proposed a unique multimatrix metabolomics fingerprinting strategy for feces, plasma, and urine using a single analytical platform (RP-UHPLC-MS). They found 9672, 9647, and 6122 features in feces, urine, and plasma, respectively, proving this approach was perfectly fitted to the concept of systems biology.

Other studies have been carried out using a highly targeted approach on dozens of molecules [24,38,39,42]. However, the major limitation of this two-metabolic phenotyping is the lack of metabolite identification, hindering its use as in disease-related biomarker detection or for potential pathway elucidation. Yet, in order to elucidate intricate metabolic pathways, the identification of the features is needed. 

Zhao et al. [24] in 2017 carried out a targeted multimatrix approach in GC-MS and were able to identify 92, 103, and 118 metabolites in serum, urine, and feces, respectively. Nevertheless, analysis of these fluids by other analytical techniques can help to widen human metabolome coverage and better understand the biological system, to provide statistically robust models and help in the discovery of biomarkers [47].

We proposed a multimatrix and multiplatform approach to achieve a high metabolome coverage with structural identification of metabolites (level 2) [48] with a high reproducibility. This identification allowed us to generate a metabolic map, collecting information from different biological compartments. This metabolic phenotype allows for a better characterization of the metabolic status of an individual and opens avenues to better understand the exchanges between different compartments.

## 3. Materials and Methods

### 3.1. Sample Preparations

All chemicals were bought from Sigma Aldrich (Saint-Quentin Fallavier, France) unless otherwise specified.

Saliva, feces, serum, and urine samples were obtained from healthy volunteers (with no dietary restrictions or antibiotic treatment prior to sample donation). Volunteers were informed and agreed with the use of their samples for this study.

#### 3.1.1. Urine

Morning-fasted urine were pooled and centrifuged (10 min at 4 °C and 15,000× *g*). Supernatants were collected in 20 aliquots of 200 µL (n = 5 per sample preparation). Three extraction protocols were tested either with solvent extraction (Urine/MeOH(1:8)), (Urine/ACN(1:8)) or urine dilution (Urine/D_2_O buffer (1:2) for NMR or Urine/deionized H_2_O (1:10) for LC-MS). For extraction protocols, 1600 µL of solvent were added to urine, vortexed for 10 min, and centrifuged (10 min at 4 °C and 15,000× *g*). Supernatants were collected (750 µL for NMR analysis and 375 µL for RP/HILIC analysis), dried under SpeedVac (ThermoFisher, Villebon sur Yvette, France), and stored at −20 °C until use. A graphical presentation of all sample preparation protocols is given in Appendix A.

#### 3.1.2. Blood

A pool of sera was divided into 30 aliquots of 200 µL. Six extraction procedures were tested (n = 5 per procedure): blood/ACN (1:2), blood/ACN (1:8), blood/MeOH (1:2), blood/MeOH (1:8), blood/MeOH:CHCl_3_ (1:1:1), and blood/MeOH/CHCl_3_ (1:1.5:2.5). Samples were vortexed, kept at −20 °C for 30 min, then centrifuged (10 min, 4 °C, 15,000× *g*). Supernatants were collected in glass tubes and divided into 3 aliquots (750 µL for RMN, 375 µL for RP-LC-MS, and 375 µL for HILIC-LC-MS) for further solvent evaporation in a SpeedVac (ThermoFisher, Villebon sur Yvette, France). A graphical visualization of all sample preparation protocols is given in Appendix A.

#### 3.1.3. Saliva

Saliva samples were pooled and 20 aliquots of 400 µL (n = 5 per solvent) were collected. For metabolite extraction, four protocols were tested (Saliva/MeOH (1:1) and (1:2), Saliva/ACN (1:1) and (1:2)), vortexed for 10 min, and centrifuged (10 min at 4 °C, 15,000× *g*). Supernatants were collected and divided into 3 aliquots (350 µL for the NMR analysis and 175 µL for the RP/HILIC analysis for the 1:1 ratio and 700 µL for the NMR analysis and 200 µL for the RP/HILIC analysis for the 1:2 ratio). Samples were dried under SpeedVac (ThermoFisher, Villebon sur Yvette, France). A graphical visualization of all sample preparation protocols is given in Appendix A.

#### 3.1.4. Feces

For better reproducibility, fecal samples were freeze-dried (FreeZone^®^4.5L, Labconco, Kansas City, MO, USA) at −107 °C and 0.2 mbar for 24 h and pooled. Sample aliquots of 6 mg (n = 5 per solvent) were added with 1 mL of the extraction solvent (water(1), MeOH/H_2_O (1:1), MeOH/H_2_O (4:1), ACN/H_2_O (1:1), ACN/H_2_O (4:1), MeOH/H_2_O/ACN (1:1:1)), vortexed for 10 min, and centrifuged (10 min at 4 °C, 15,000× *g*). Supernatants were collected in 3 aliquots (450 µL for the NMR analysis and 225 µL for the C18/HILIC analysis) and dried under SpeedVac (ThermoFisher, Villebon sur Yvette, France) or freeze-dried for H_2_O solvent. A graphical visualization of all sample preparation protocols is given in Appendix A.

Dried residues were dissolved in 100 µL of MeOH/H_2_O (1:9) for RP-LC and in 100 µL of ACN/H_2_O (75:25) for HILIC. Dried residues were dissolved in 200 μL of a deuterated buffer (0.2 M potassium phosphate buffered deuterium oxide (pH = 7.44 ± 0.5) and 10 μL of deuterium oxide (D_2_O) with external reference [3-trimethylsilylpropionic acid] (TSP) at 3.2 mM for the ^1^H-NMR analysis.

### 3.2. Data Acquisition

#### 3.2.1. UHPLC-MS

As previously described [49], LC-HRMS analysis was performed on an UPLC Ultimate WPS-3000 system (Dionex, Germany), coupled to a QExactive mass spectrometer (Thermo Fisher Scientific, Bremen, Germany).

The chromatography system was equipped separately with two columns: RP-LC (Reverse Phase Liquid Chromatography) Phenomenex Kinetex^®^ XB-C18 (1.7 µm 100 A 150x 2.1 mm from Phenomenex, CA, USA) kept at a constant temperature of 55 °C. The solvent system comprised mobile phase A (ultrapure water) and mobile phase B (methanol), both acidified with 0.1% formic acid. A constant flow rate of 0.4 mL/min over a run of 24 min was used with multistep gradient. The multistep gradient (followed by 2 min equilibration time) was programmed as follows: 0–6 min, 0.1–25% B; 6–10 min, 25–80% B; 10–12 min, 80–90% B; 12–21 min, 90–99.9% B; 21–23 min, 99.9% B; 23–24 min, 99.9–0.1% B. 

HILIC (Hydrophilic Interaction Liquid Chromatography) Waters Cortecs^®^ (unbon-ded silica; 1.6 µm 100 A 150x2.1mm from Waters, Ireland) kept at a constant temperature of 40 °C. The solvent system comprised mobile phase A (ultrapure water; pH = 3.0) and mobile phase B (acetonitrile added with 5% water to dissolve ammonia formate and prevent precipitation) both contain 10mM of ammonium formate and acidified with 0.5% formic acid. A constant flow rate of 0.3 mL/min over a run of 27 min was used with a multistep gradient. The multistep gradient (followed by 2 min equilibration time) was programmed as follows: 0–8 min, 5–18% A; 8–15 min, 18–25% A; 15–20.5 min, 25–75% A; 20.5–22 min, 75–97% A; 22–23 min, 97–5% A; 23–27 min, 5% A. 

The auto sampler temperature (Ultimate WPS-3000 UHPLC system, Dionex, Germany) was set at 4 °C.

HESI (head electrospray ionization) source was used for both chromatography systems with a spray voltage of 3.5 kV, a capillary temperature of 325 °C, a heater temperature of 350 °C, a sheath gas flow of 50 arbitrary units (AU), an auxiliary gas flow of 13 AU, a sweep gas flow of 3 AU, and a tube lens S-Lens RF level 50, operated in positive (ESI+) and negative (ESI−) electrospray ionization modes (one run for each mode).

Detection was performed with a full-scan acquisition at 70,000 resolution (*m*/*z* = 200), which ranged from 58.0 to 870.0 *m*/*z*, with an automatic gain control (AGC) target of 105 charges and a maximum injection time (IT) of 250 ms.

Xcalibur 2.2 software (Thermo Fisher Scientific, Bremen, Germany) controlled the system.

#### 3.2.2. NMR

^1^H-NMR spectra were acquired at 298K on an AVANCE III HD 600 MHz system (Bruker Biospin, Karlsruhe, Germany) equipped with a Bruker 5 mm TCI CryoProbe with Z-gradient. ^1^H-NMR spectra were recorded with the «noesypr1d» pulse sequence with a relaxation delay of 20 s, on a sweep width of 12 ppm, 64 k data points, an acquisition time of 4.55 s, with 64 transients, and 8 dummy scans.

### 3.3. Data Processing

#### 3.3.1. UHPLC-MS

Data were analyzed using Workflow4Metabolomics (W4M), an online platform for metabolomics data pre-processing including XCMS Online [50,51]. A pre-processing workflow was created following three global steps: (i) the extraction method used CentWave-chromatographic peak detection with a max tolerated ppm *m*/*z* deviation of 15.0 and a minimum difference in *m*/*z* of 0.008, (ii) the alignment method was completed by grouping chromatographic peaks within and between samples by PeakDensity with a bandwidth of 10 and by using Obiwarp for the retention time correction [52], and (iii) Camera.annotate was used to annotate isotope peaks, adducts, and fragments in the peak list in order to keep only the monoisotopic peak. Detected peaks were identified by the module bank_inhouse with an internal database made by Mass Spectrometry Metabolite Library of Standards MSMLS^®^ (IROA Technologies™). Only metabolites detected in all repetitions were kept for the further analysis.

#### 3.3.2. ^1^H-NMR

Spectra were processed using TopSpin 3.6 software (Bruker Biospin, Karlsruhe, Germany) and were reduced to buckets of variable widths using AMIX software (Analysis of MIXture, version 3.2, Bruker Karlsruhe, Germany). Bucket identification was done using ChenomX software (NMR Suite 7.7, ChenomX, free software, Edmonton, Canada), an in-house database, and literature [22].

### 3.4. Data Fusion

Once the pre-processing of the RP and HILIC MS data were completed, Workflow4Metabolomics generated a compound list for RP-LC-MS and for HILIC-LC-MS. ChenomX generated the NMR data list. The three compound lists were combined thanks to the identification of peaks. When metabolites were detected by several platforms, and to keep the highest data quality, a data cleaning was done according to the best replicate chosen by the lowest coefficient of variation (CV%) to get rid of the redundancy.

### 3.5. Data Analysis

Data obtained from the different preparation protocols for each matrix and the different analytical tools were analyzed. The number of metabolites, unique metabolites, and the coefficient of variation (CV%) were calculated.

To identify the impacted metabolic pathways using the extracted metabolites, we analyzed the set of metabolites for each preparation by the Network Explorer module [53] implemented in Metaboanalyst4.0, a freely available web-based metabolomics analysis suite (MetaboAnalyst4.0, Quebec, QC, Canada http://www.metaboanalyst.ca/ (accessed on 10 March 2020)) [53]. The Network Explorer module was developed to support a network-based approach for multi-omics data integration and interpretation. This module is used to build the mapping of metabolites onto different types of metabolic pathways. The results are presented in a table that summarizes the most significant metabolite sets with their *p*-values in the Network Explorer. A cutoff was chosen at a raw *p*-value of 0.05 to retain the most significantly enriched metabolic pathways. 

In this study, for each biofluid analyzed, we selected the extraction method based on the number of total metabolites, their CVs, the number of unique metabolites, and the predicted number of metabolic pathways found by using MetaboAnalyst 4.0.

To visualize the metabolic coverage using the four matrices, we used an interactive pathway explorer (iPath) (https://pathways.embl.de/ accessed on 5 November 2020) [54].

Complementarity of matrices was evaluated using an upset graph (https://bioinfo-fr.net/contrarie-par-les-diagrammes-de-venn-decouvrez-les-diagrammes-upset/ (accessed on 7 April 2020)) [46].

### 3.6. Intraday Precision

Validation method parameters of untargeted metabolomics [55] are specific but intraday precision needs to be addressed. Precision is a crucial point to validate [56]. This parameter was measured by injecting a series of samples from multiple preparations coming from the same and homogeneous pool. It allows for the evaluation of the closeness of repeated measures of the same sample/analyte [55]. The pools previously prepared for each matrix were thus extracted 16 times to validate the reproducibility of the method. The 16 extractions (named repeated samples) were analyzed in LC-MS and NMR. Previous metabolites detected for each matrix were researched in repeated samples. The variation coefficient was calculated for each metabolite in order to validate the reproducibility of the workflow.

## 4. Conclusions

This study presents the development of a multimatrix and multiplatform analytical method. This approach makes it possible to see the complementarity between the matrices but also between the platforms. It therefore allows a broad description of the human metabolome. It makes it possible to set up a methodology for obtaining a metabolomic map that describes an individual at a specific time. This methodology can therefore be applied in clinical studies comparing two or more groups with different metabolomes. Moreover, a statistical study of the metabolic interaction between metabolites from different compartments could help to understand exchanges. This approach needs to be implemented with an absolute quantification detection in order to follow an individual along his lifetime. 

The aims of this work were to explore and represent in a robust way the whole metabolome in the different compartments of the same individual. The multiplicity of data obtained will provide important information on the state of health of a person, both by comparing their metabolomes over time or with a reference population, and by identifying mechanisms for understanding exchanges in compartments. This multilayer and multiplatform metabolic phenotype has the potential to detect very early metabolic trajectory drifts associated with a pathology.

## Figures and Tables

**Figure 1 molecules-26-04111-f001:**
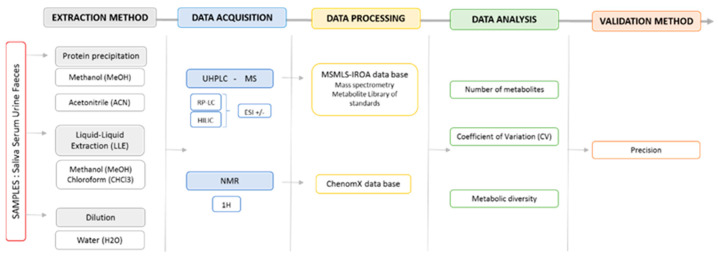
Schematic presentation of the global workflow of preparation procedures applied to urine, saliva, blood and feces pools for the selection of the optimum protocol for the maximum metabolome coverage by LC-MS and ^1^H-NMR.

**Figure 2 molecules-26-04111-f002:**
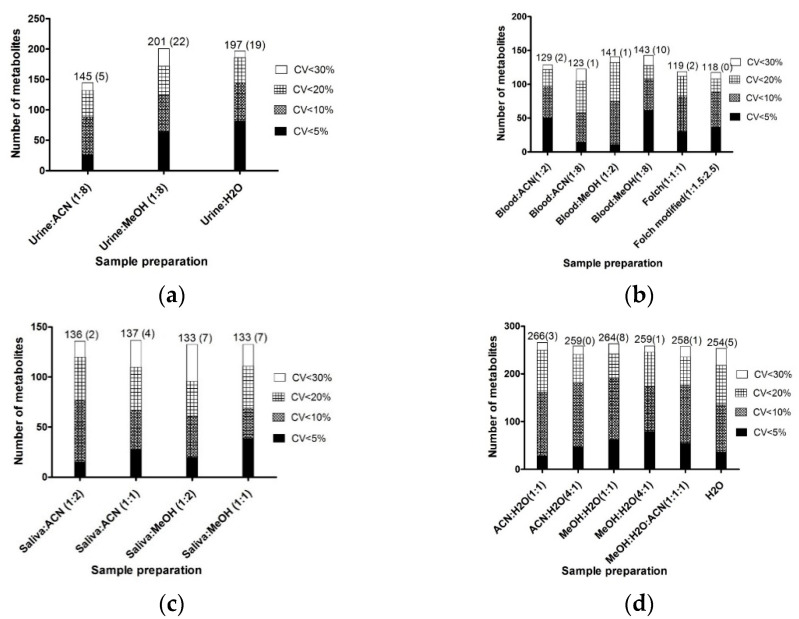
Histogram showing the total number of metabolites extracted for each preparation and their respective reproducibility evaluated by CV (coefficient of variation) in urine (**a**), blood (**b**), saliva (**c**), and feces (**d**). The numbers of metabolites found to be unique from each preparation condition are in brackets.

**Figure 3 molecules-26-04111-f003:**
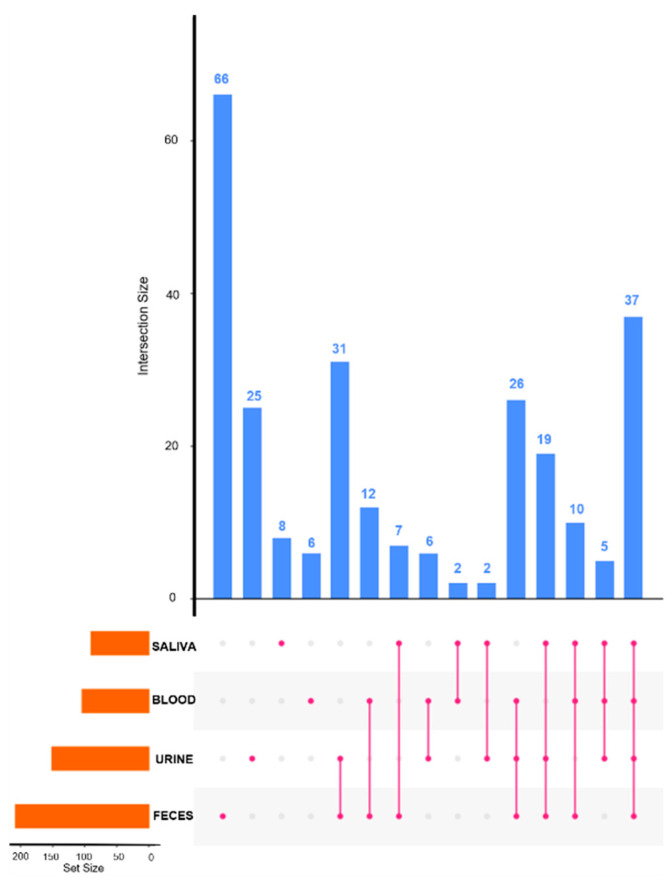
Upset diagram to show the complementarity in term of metabolite coverage across the different biological matrices (Set Size = number of total compounds per matrix, Intersection Size = number of compounds detected in different matrices).

**Figure 4 molecules-26-04111-f004:**
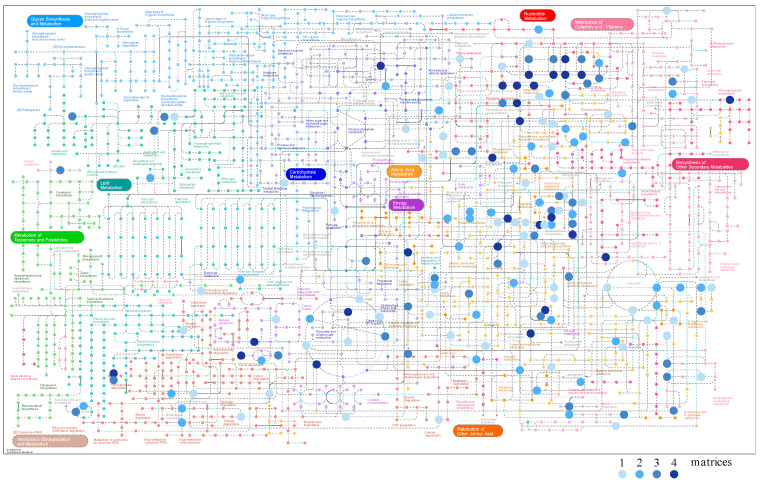
Metabolomic card showing metabolites from urine, blood, saliva, and feces through the metabolomics pathway to illustrate an individual. Metabolites expressed in one, two, three, or four matrices are shown in the blue gradient color.

**Table 1 molecules-26-04111-t001:** Numbers of metabolites recovered in repeated samples compared to metabolites found in the chosen extraction protocol. Variation coefficients (CVmean) and (CVmin–Cvmax) were calculated from the fusion of the three platforms.

	Urine	Blood	Saliva	Feces
Sample preparation chosen	Urine/MeOH (1:8)	Blood/MeOH (1:8)	Saliva/ACN (1:2)	MeOH/H_2_O (1:1)
Number of metabolites	151	104	90	208
CV mean % (min–max)	7% (2–53%)	8% (3–53%)	7% (2–26%)	11% (3–40%)

## Data Availability

Data are contained on the Appendix A.

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
