# Peer review of "Optimization of Sample Preparation for Metabolomics Exploration of Urine, Feces, Blood and Saliva in Humans Using Combined NMR and UHPLC-HRMS Platforms"

_molecules, 2021, doi:10.3390/molecules26144111_

Round 1

Reviewer 1 Report

This study compared the reproducibility and metabolome coverage of samples extracted by different solvents. Overall, the manuscript is well prepared, but the authors should check the manuscript carefully to correct grammar errors and typos.

For instance,

Line 54 “is the most widely applied technologies”, should be “is one of the most widely applied technologies”

Line 78 “Another concern is sample preparations ”, should be “Another concern is sample preparation”

Line 79 “biofluid” should be “biofluids”

Line 86 “an unique” should be “a unique”. Line 95 has the same problem.

Some other comments:

  1. A most common blood handling method is using 3 folds of blood volume methanol to precipitate protein. The author only compared 1:2 and 1:8 blood: methanol. I am worried about the high content of methanol will cause hydrophilic metabolites loss. Why not also evaluate 1:3 or 1:4?
  2. Line 448. Did the author dissolve ammonia formate in pure ACN? Usually small percent of water (5%~10%) needs to be added into ACN to help ammonia formate dissolving and prevent precipitation. The authors should confirm the LC-MS conditions. What is the final pH of mobile phase A and B? That is very important in HILIC analysis.

Author Response

We thank reviewer for his advices and comments.

Please find in the attached file our answers and modifications.

Best regards

Reviewer 2 Report

Manuscript deals with the optimization of biological sample preparation for metabolomics exploration. Authrs analysed urine, feces, blood and saliva samples using NMR and UHPLC-HRMS. 

It is very good work, however some points have to be corrected:

Figure 1 According to data acquisition, I do not understand the idea to divide chromatographic separations to C18 and HILIC. It Should be RP-LC and HILIC. C18 is the name of the column, not the mode of the separation. HILIC also uses some columns. It has to be also stated in lines 93-95, 108-109, etc. It is important from the analytical point of view.

The term: analytical method is in my opinion misunderstood in Fig. 1. 

Mass spectrometry rather than mass spectroscopy! (over the whole text)

What type of the column was used for HILIC separation? It looks like HILIC is the name of the column in this work!!!

Usually, the manufacture is provided to the column.

What were the gradients profiles in RP and HILIC?

Author Response

(The authors gave the same response as above.)
